# Molecular Evolution and Phylogeography of the Crimean–Congo Hemorrhagic Fever Virus

**DOI:** 10.3390/v17081054

**Published:** 2025-07-28

**Authors:** Paula Iglesias-Rivas, Luis Daniel González-Vázquez, Miguel Arenas

**Affiliations:** 1CINBIO, Universidade de Vigo, 36310 Vigo, Spain; paula.iglesias.rivas@uvigo.es (P.I.-R.); luisdaniel.gonzalez@uvigo.es (L.D.G.-V.); 2Department of Biochemistry, Genetics and Immunology, Universidade de Vigo, 36310 Vigo, Spain

**Keywords:** molecular evolution, CCHFV, recombination, molecular adaptation, rate of evolution, genetic diversity, phylogeography

## Abstract

The Crimean–Congo hemorrhagic fever virus (CCHFV) is a single-stranded, segmented RNA virus belonging to the *Nairoviridae* family, and it is rapidly expanding across Africa, Asia, and southern Europe, probably favored by climate change and livestock trade. Its fatality rate in humans reaches up to 40%, and there is currently no specific treatment or vaccine available. Therefore, the development of therapies against CCHFV is essential, and their design requires understanding of the molecular evolution and genetic distribution of the virus. Motivated by these concerns, we present a comprehensive review of the molecular evolution, genetic characterization, and phylogeography of CCHFV, and we discuss their potential implications for therapeutic design. Specifically, we describe the virus’s capacity to increase its genetic diversity through numerous mutations, recombination events, and genomic reassortments, which affect fundamental viral functions such as RNA binding, host–virus interactions, viral entry, and polymerase activity. We also assess the presence of temporal heterogeneous rates of evolution and molecular adaptation among CCHFV coding regions, where purifying selection is generally predominant but diversifying selection is observed in molecular regions associated with host adaptation and transmission. We emphasize the importance of understanding the complex molecular evolution of CCHFV for the rational design of therapies and highlight the need for efforts in surveillance, evolutionary prediction, and therapeutic development.

## 1. Introduction

The Crimean–Congo hemorrhagic fever virus (CCHFV) belongs to the *Orthonairovirus* genus of the *Nairoviridae* family and is one of the most virulent tick-borne viruses. It poses a threat to both humans and animals [1,2], with a high mortality rate that can reach up to 40% [3]—in some outbreaks, up to 80% [4]—due to the zoonotic disease that it causes—Crimean–Congo hemorrhagic fever (CCHF). There are currently no vaccines or specific antiviral treatments available for this emerging pathogen, and the World Health Organization (WHO) has listed CCHF among its top-priority diseases in public health emergency contexts [5]. Therefore, the development of effective therapies against this virus is of critical importance, and a thorough understanding of its replication mechanisms and molecular evolution is essential to support it.

CCHFV is usually transmitted by tick bites or through contact with bodily fluids from infected humans or animals [6,7]. Ticks become infected when feeding on infected domestic or wild animals, including ungulates, small mammals, and even migratory birds capable of transporting the virus over long distances [1,2,8,9]. Although CCHFV usually circulates through natural (enzootic) tick–vertebrate–tick transmission cycles, horizontal (co-feeding, venereal transmission, trans-stadial) and vertical (transovarial) transmission among ticks has also been observed [1,10,11]. Regarding its current geographic distribution, CCHFV is endemic in Africa, the Balkans, the Middle East, and Asia, although outbreaks have also been reported in other regions (Figure 1) [12]. For instance, its recurrent detection in countries such as Spain raises questions about circulation in neighboring countries, including Portugal, Italy, and France [5,13,14].

At the molecular level, CCHFV presents a negative-sense single-stranded RNA genome of approximately 19.2 kb in length, consisting of three segments: small (S; approximately 1.7 kb), medium (M; approximately 5.4 kb), and large (L; approximately 12.1 kb) (Figure 2 and Table 1). These genomic segments encode a variety of proteins essential for viral transmission and the replication cycle (details are provided in the next section). CCHFV exhibits a highly active evolutionary process, involving not only mutations but also genetic recombination, in which gene segments are exchanged between co-infecting parental viruses within a single host cell [17], leading to the emergence of novel viral strains with particular genomic compositions [18,19]. Mutation and recombination can significantly increase viral genetic variability, providing the raw material upon which selection operates, and are crucial for enabling the virus to adapt to its hosts [20]. This remarkable molecular evolution has also supported the development of phylogenetic classifications. For example (with further details provided in a later section), phylogenetic analyses of the S segment classified CCHFV into six major geographic clades or lineages: three predominantly found in Africa (lineages I–III), two in Europe (lineages V and VI), and one in Asia (lineage IV) (Figure 3) [21]. Indeed, seven and six lineages were used to classify the M and L segments, respectively (Figure 3). Some studies have proposed a more complex classification based on genotypes associated with geographic regions [22,23,24], considering that a lineage can include more than one genotype. For example, seven genotypes were proposed for the S segment, namely, “Africa 1”, “Africa 2”, “Africa 3”, “Asia 1”, “Asia 2”, “Europe 1”, and “Europe 2”, and additional genotypes were proposed for the M and L segments (Figure 3). This heterogeneity among genomic segments suggests distinct evolutionary processes and is also associated with differences in pathogenic potential. For example, the “Europe 1” genotype caused more clinical cases than the “Europe 2” genotype; thus, virulence can vary among genotypes [25,26]. The observed broad geographic distribution of this virus can be attributed to several factors, including the introduction of ticks into new territories (perhaps facilitated by climate change), the migration of infected animals, and transmission from infected humans [27,28,29].

**Table 1 viruses-17-01054-t001:** Genomic regions, proteins, and associated functions in CCHFV. For each genomic segment, the table indicates the corresponding genomic regions and proteins, along with their respective locations and main functions. References supporting the described protein functions are also provided.

Genomic Segment	Genomic Region		Location (Amino Acids)	Function	References
S	Nucleoprotein (NP)		19-502	Virus replication	[30,33]
	Non-structural (NSs)		Overlaps with NP and varies among strains	Apoptosis induction and mitochondrial membrane potential disruption	[30,34]
M	Glycoprotein precursor (GPC)	Mucin-like domain (MLD)	19-243	Virus infectivity, immune evasion, and tropism	[30,35]
		Glycoprotein 38 (GP38)	244-519	Virus infectivity, immune evasion, viral spread, and tropism	[35,36,37,38]
		Glycoprotein N (Gn)	520-842	Receptor binding, virion assembly	[35,36,39]
		Non-structural (NSm)	843-1040	Virion assembly and budding	[30,35,40]
		Glycoprotein C (Gc)	1041-1684	Membrane fusion	[35,36]
L	RNA-dependent RNA polymerase (RdRP)		25-3970	Viral genome, transcription, and replication	[30,32,41]

Monitoring and understanding the molecular evolution of CCHFV across its diverse lineages can provide insights into how the virus adapts to different environments and hosts and may help predict the emergence of new viral variants. Indeed, this knowledge could help identify molecular targets for surveillance, diagnostic methods, and potential treatments, which are demanded for managing infections and for anticipating and mitigating future outbreaks. We present a comprehensive review of the current knowledge on the molecular evolution and genetic distribution of CCHFV and how this information could be useful in designing diagnostic methods and potential therapies.

## 2. The Genomic Structure and Proteins of CCHFV

As previously noted, the three genomic segments of CCHFV encode a variety of proteins, including eight structural, non-structural, and accessory proteins, that are involved in host recognition, virus entry and uncoating, replication, transcription, assembly, and release, among other functions (Figure 2 and Table 1). Key viral proteins are the spike glycoproteins Gn and Gc (encoded by the M segment), located on the lipid envelope, which are responsible for receptor binding and viral entry [6,39,42]. Additionally, the nucleoprotein (NP, encoded by the S segment) and RNA-dependent RNA polymerase (RdRP or L protein, encoded by the L segment) form ribonucleoprotein (RNP) complexes [30,42], which encapsulate the genomic segments and mediate viral replication. The specific coding regions and the corresponding proteins encoded by each genomic segment are detailed below.

### 2.1. The S Genome Segment

The small genome segment encodes the nucleocapsid protein (NP) and the non-structural S protein (NSs). The NP has a globular “head” domain and a flexible “arm” domain that form super-helical structures, enabling it to bind and package the viral RNA genome into RNP complexes. These complexes are essential for genome protection, replication, and transcription [30,43]. NP also interacts with host factors such as GTPase MxA (i.e., with a conserved caspase-3 cleavage site), indicating a role in modulating host–virus interactions [44]. It may also influence cellular processes such as apoptosis and immune evasion (i.e., it includes an antigenic region at amino acid positions 235–305, which induces immune responses in diverse hosts [45,46]). In addition, several studies showed that NP performs additional activities because it is released from cells even in the absence of other viral components, exhibits self-assembling properties, and generates bunyavirus-like spherical particles in viral budding [30,33,47]. These multifunctional properties highlight NP as a key factor in viral replication, viral mRNA translation [48], and host adaptation [42,49,50]. In contrast, the NSs protein is less characterized. It has been shown to induce apoptosis of the infected cells by disrupting the mitochondrial membrane potential [34].

### 2.2. The M Genome Segment

The medium genomic segment encodes a glycoprotein precursor (GPC) that undergoes proteolytic processing to generate two precursor intermediates, preGn and preGC, which subsequently mature into the two major envelope transmembrane glycoproteins, Gn and Gc. This segment also encodes several accessory proteins, including the glycoprotein GP38 and the small non-structural protein NSm [30,31,33,35,40]. Notably, PreGn contains two cleavage sites that are essential for virus infectivity: (i) a conserved furin cleavage site (RSKR^247^ motif), unique to CCHFV, located between the mucin-like domain (MLD) and the GP38 region; and (ii) a site-1 protease (S1P) cleavage site (^516^RRLL^519^) at the GP38-Gn interface. The latter protease, also known as subtilisin kexin isozyme-1 (SKI-1), processes the viral GPC into mature Gn and Gc and is also crucial for incorporating viral glycoproteins into the virions [40,51,52]. Next, GP85 and GP160 glycoproteins arise from glycosylation of the MLD-GP38, accompanied by partial cleavage at the RSKR^247^ motif [53]. These glycoproteins, along with GP38, are secreted by infected cells and targeted to the plasma membrane and viral envelope, suggesting roles in viral dissemination [38]. Also, GP38 and MLD are implicated in virus infectivity, immune evasion, and cell tropism [30,36]. Specifically, GP38 facilitates proper folding and hetero-oligomerization between the M segment and structural proteins [40] and also contributes to virion assembly [51]. In fact, some authors demonstrated that GP38 is essential for producing infectious particles [40]. Reduced GP38 expression impairs preGn-to-Gn conversion and slows viral replication [52]. Regarding Gn and Gc, they are transmembrane proteins embedded in the viral envelope and are responsible for receptor binding, membrane fusion, and virion assembly [6,35,36]. Gn facilitates virion anchoring and interacts with RNP complexes. Notably, specific domains in Gn (such as zinc finger motifs) are essential for genome packaging and viral assembly [39]. Gc mediates fusion between the viral and host cell membranes. Mutations or structural alterations in Gc can influence viral tropism, entry efficiency, and antigenic properties [35,36]. Although less characterized, the NSm protein appears to contribute to virion assembly and to virus–host interactions [30,35,40].

### 2.3. The L Genome Segment

The large segment encodes the RNA-dependent RNA polymerase protein (RdRp), which is essential for viral RNA synthesis during replication as well as for the transcription of the viral genome into mRNA [6,30]. This protein also contains several domains and motifs (i.e., OTU-like protease domain, leucine zipper, and zinc finger) that contribute to immune evasion by inhibiting interferon signaling pathways [6,30,41,54].

## 3. Evolutionary Mechanisms of CCHFV

In general, CCHFV exhibits high genetic diversity, driven by several evolutionary mechanisms: (i) mutations introduced by its error-prone RNA polymerase [55,56]; (ii) recombination events that can occur during coinfection of the same host cell [39,56,57,58,59]; (iii) and additional genomic reassortments among segments [17,19,56,59], which are common processes in segmented RNA viruses. Together, these mechanisms generate viral variants upon which selection acts, favoring and fixing those better adapted to the environment (i.e., the host immune system). These adapted variants can persist and spread through different vectors and hosts [6,33,60,61,62]. An important aspect of CCHFV evolution is the balance between genetic variation and conservation, which is critical for viral replication, pathogenesis, and transmission [6]. For example, the M segment, which, as noted, encodes envelope glycoproteins, undergoes evolutionary pressures due to its involvement in host interactions and induction of neutralizing antibodies [33,61]. In contrast, the S segment, which, as noted, encodes the nucleocapsid protein, can accumulate mutations that affect RNA binding, virion assembly, or immune evasion [31,63]. In the following sections, we examine in detail the evolutionary mechanisms operating in CCHFV, highlighting their specific characteristics and functional consequences.

### 3.1. Genetic Diversity and the Rapid Mutation Process of CCHFV

RNA viruses lack proofreading activity in their RdRP, which enables rapid mutation and increased genetic diversity, especially under conditions of fast replication rates [64]. However, arthropod-borne RNA viruses often exhibit lower genetic diversity compared to other RNA viruses due to selective constraints imposed by the need to maintain fitness in both arthropod and vertebrate hosts [11,55]. CCHFV is an exception, exhibiting notably high variability across all three of its genomic segments. In particular, the observed nucleotide diversity is approximately 20%, 31%, and 22% in the S, M, and L segments, respectively. At the protein level, the observed amino acid diversity is approximately 8% in the NP, 27% in the GPC, and 10% in the L protein [11,65,66]. Geographic variation of genetic diversity is also notable, with the S segment ranging from 10% to 20% and the M segment reaching up to 31% in some epidemic areas [37]. At the protein level, the S and L segments generally show over 95% amino acid conservation across strains, whereas the M segment can show less than 75% amino acid identity between distantly related strains [33,60]. Interestingly, among regions of the M segment, MLD usually shows the highest sequence variability [67]. The glycoproteins Gn and Gc generally show high diversity, although lower than that of MLD and higher than the diversity generally observed in the NS protein [68,69]. In general, these differences in genetic variability reflect distinct evolutionary pressures acting on each segment, which are further examined in the following sections.

In general, CCHFV can accommodate mutations in key viral proteins, which can lead to changes in virus–host interactions, replication efficiency, and virion assembly [31,63]. Table 2, Table 3 and Table 4 indicate relevant characterized mutations in CCHFV proteins, organized by genomic segment. Below, we highlight the most functionally significant cases. In proteins encoded by the S genomic segment, we highlight specific amino acid changes in the NP at RNA binding sites (such as K132A, Q300A, and K411A) that were associated with disruptions in viral transcription and replication, generally affecting the replication process (Table 2). In proteins encoded by the M genomic segment, the MLD of the GPC stands out for harboring a particularly high number of mutations—more than 20 were reported (though not individually annotated by the authors) [31]—which may facilitate immune evasion and adaptation to different hosts [11,31,68,70]. Additionally, specific mutations in the envelope glycoproteins Gn and Gc, such as T1045I and A1046V, were associated with disruption of hydrophobic interactions, while others like I778T may alter protein folding (Table 3). Regarding proteins encoded by the L genomic segment, several mutations occurred near the catalytic site of the RdRP (such as V2074I and I2134T/A), which can influence polymerase activity and solubility (Table 4). Altogether, these findings underscore the capacity of CCHFV to accumulate mutations, which serve as a primary source of virus molecular evolution and enable adaptation to the environment.

**Table 2 viruses-17-01054-t002:** Main mutations observed in proteins encoded by the CCHFV S genomic segment. For each observed mutation (row), the table indicates its position and the corresponding amino acid change, the reported influence on the functions of the overlapping NP and NSs proteins, and associated references.

NP and NSs Proteins		
Mutation: Exchanged Amino Acids and Protein Site	Influence on Protein Function	References
R15K	Can lead to loss of external interactions	[31]
K90A	Reduces replicon activity and can play a role in RNA binding	[42]
K91A	Apparently without effects on protein function	[42]
K98A	Apparently without effects on protein function	[42]
E112A	Apparently without effects on protein function	[42]
T124A/S	Can affect hydrophobicity and loss of external interactions	[31]
G125N	Can produce loss of protein function, as glycine could allow protein flexibility	[31]
K132A	Removes the replicon activity	[42]
R140A	Apparently without effects on protein function	[42]
S149A	Apparently without effects on protein function	[42]
H195R	Can affect intramolecular interactions with the viral genome	[31]
I228M	Since it is in the arm domain, it could affect protein–protein interactions. It could potentially increase virulence in mice	[63]
K251E	Since it is in the arm domain, it could affect protein–protein interactions. It could potentially increase virulence in mice	[63]
I246V	Can create a cavity in the protein core	[31]
Q300A	Removes the replicon activity	[42]
S301G	Can cause loss of external interaction and loss of hydrophobicity and create empty space in the core of the NP	[31]
K342A	Apparently without effects on protein function	[42]
K343A	Apparently without effects on protein function	[42]
K411A	Contributes to DNase activity of the CCHFV N protein. Direct role in CCHFV gene expression	[42]
V436I	New residue is bigger and does not fit in the protein core	[31]
H453A	Apparently without effects on protein function	[42]
H456A	Can play a role in RNA binding	[42]
Y470A	Apparently without effects on protein function	[42]

**Table 3 viruses-17-01054-t003:** Main mutations observed in proteins encoded by the CCHFV M genomic segment. For each observed mutation (row) at each protein, the table indicates its position and the corresponding amino acid change, the reported influence on the protein function (if known), and associated references.

Signal Peptide (Located Before GPC at N-Terminal)		
Mutation: Exchanged Amino Acids and Protein Site	Influence on Protein Function	References
I9V	Can affect processing and subsequent maturation of viral glycoproteins	[31]
**Glycoprotein N**		
**Mutation: Exchanged amino acids and protein site**	**Influence on protein activity**	**References**
P523S/T/F	Can affect Gn intramolecular interactions	[31]
R579K	Can affect Gn intramolecular interactions	[31]
N592S	Can affect Gn intramolecular interactions	[31]
V718F/A/I/L	Can affect Gn intramolecular interactions	[31]
L725F	Can affect Gn intramolecular interactions	[31]
I778T	Can generate loss of intramolecular interactions and affect Gn protein solubility and stability	[31]
**Glycoprotein C**		
**Mutation: Exchanged amino acids and protein site**	**Influence on protein activity**	**References**
T1045I	Generates loss of hydrogen bonds and intramolecular interactions	[31]
A1046V	Generates loss of hydrogen bonds and intramolecular interactions	[31]
G1158E	Affects protein folding and local protein structure	[31]
L1331L	Synonymous mutation without phenotypic effects	[63]
A1451T	Generates loss of hydrophobic interactions	[31]
H1527Y	Generates loss of hydrophobic interactions	[31]
M1597I	Generates loss of intramolecular interactions	[31]
K1652R	Generates bumps in the protein structure	[31]
**Glycoprotein 38**		
**Mutation: Exchanged amino acids and protein site**	**Influence on protein activity**	**References**
G250D/N/E	Can affect GP38 intramolecular interactions	[31]
Q273H/R/D	Can affect GP38 intramolecular interactions	[31]
V385A/D/T	Can affect GP38 intramolecular interactions	[31]
R475R	Synonymous mutation without phenotypic effects	[63]
D484N	Can affect GP38 intramolecular interactions	[31]
**Non-structural M**		
**Mutation: Exchanged amino acids and protein site**	**Influence on protein activity**	**References**
Q844H	Can affect NSm intramolecular interactions	[31]
C865Y	Located in the NSm protein but without a precisely described function	[63]
K944T/A	Can affect NSm intramolecular interactions	[31]
T927A/V	Can affect NSm intramolecular interactions	[31]
L955R	Can affect NSm intramolecular interactions	[31]
K1038R	Can affect NSm intramolecular interactions	[31]

**Table 4 viruses-17-01054-t004:** Main mutations observed in proteins encoded by the CCHFV L genomic segment. For each observed mutation (row) at each protein, the table indicates its position and the corresponding amino acid change, the reported influence on the protein function (if known), and associated references.

RNA-Dependent RNA Polymerase		
Mutation: Exchanged Amino Acids and Protein Site	Influence on Protein Function	References
S2007N	Located in a region of the L protein without a precisely described function	[63]
V2074I	Can increase protein solubility and disrupt protein function, as they are located close to a highly conserved region	[71]
I2134T	Can increase protein solubility	[71]
V2148	Can increase protein solubility and disrupt protein function, as they are located close to a highly conserved region	[71]
V2686V	Synonymous mutation without phenotypic effects	[63]
Q2695H	Can decrease protein stability	[71]
P3281L	Located in a region of the L protein without a precisely described function	[63]
E3847E	Synonymous mutation without phenotypic effects	[63]

### 3.2. Recombination Is Frequent and Essential in CCHFV

In RNA viruses, recombination occurs during coinfection of the same host cell by two or more virions, leading to a reshuffling of mutations within the viral population [72]. This evolutionary mechanism generally increases viral genetic diversity and can produce variants better adapted to new environments [17,20,57,58]. Moreover, detecting and accounting for recombination is crucial to avoid biases in the various phylogenetic analyses commonly used to study virus evolution, such as phylogenetic tree reconstruction [73], ancestral sequence reconstruction [74], and detection of selection [75,76]. Therefore, when genetic signatures of recombination are present, this evolutionary force must be incorporated into the analyses [77]. Despite this knowledge, recombination is often overlooked in phylogenetic studies, and we thus recommend explicitly considering recombination to avoid the associated biases.

Although recombination is generally rare in negative-sense RNA viruses [78], multiple studies have indicated that recombination is relatively frequent in CCHFV, contributing to increased genetic diversity [17,37,66,79] and the emergence of variants with novel antigenic epitopes [57]. Recombination events were most frequently detected in the S genomic segment and, to a lesser extent, in the L genomic segment, whereas the M segment showed recombination events less frequently [17,57,59].

Genomic reassortments, in which whole genome segments are exchanged, are also common in CCHFV [57,58,79]. Like recombination, these events contribute to increased genetic diversity [57,65,80] and play a relevant role in shaping the geographic structure of CCHFV genetic variation [56,58]. Importantly, recombinant and reassorted variants with unfavorable combinations are purged by selection, and only those with favorable fitness for virus survival tend to persist in the population [18,81,82]. Regarding the observed reassortments, those involving the S and L segments tend to occur among closely related lineages, whereas reassortments of the M segment are more frequent [37] and were observed across more distantly related groups, resulting in novel molecular variants [56,83]. There are diverse biological factors that allow understanding of the importance of these evolutionary events—for instance, the functional interdependence of the NP (S segment) and RdRP (L segment), which together form the RNP complex that is essential for viral replication [30]. Selection tends to preserve the structure and function of the NP and RdRP, as alterations in these genomic regions can impair replication and genome packaging [56,84,85,86]. In particular, when NP and RdRP belong to highly divergent lineages, structural incompatibilities may arise that disrupt RNP complex assembly or stability, impair polymerase activity, and ultimately affect viral RNA replication and transcription. As a consequence, only reassortant viruses with functionally compatible S and L segments tend to persist and be observed in nature [86]. Another example involves the M genomic segment, where reassortment events tend to increase diversity in regions encoding the envelope glycoproteins, leading to the generation of novel antigens that may facilitate immune evasion and adaptation to different hosts and vectors [58,83,87].

### 3.3. The Temporal and Spatial Heterogeneous Rate of Molecular Evolution in CCHFV

The estimated rates of evolution for CCHFV range from 1 × 10^−4^ to 3 × 10^−4^ substitutions per site per year [22,39,59,61], consistent with the general rates reported for RNA viruses [55,56]. Notably, the rate of evolution in RNA viruses does not always follow a strict molecular clock [55]; instead, it can vary over time. This temporal variability can be caused by factors such as temporal changes in mutation or replication rates, population size, selective pressures and host–virus interactions, among others [55,88]. In the case of CCHFV, the molecular clock hypothesis did not fit with the observed evolutionary dynamics in any of the three genomic segments. Moreover, the rate of evolution varied among lineages and genomic regions (Table 5). Actually, using a relaxed molecular clock model, the estimated rates of evolution were 1.09 × 10^−4^, 1.52 × 10^−4^, and 0.58 × 10^−4^ substitutions per site per year for the S, M, and L genomic segments, respectively [22]. Interestingly, certain subregions within the S segment exhibited higher rates of evolution than the entire segment, indicating spatial heterogeneity of the rate of evolution across the genome [22,39,55].

**Table 5 viruses-17-01054-t005:** The estimated rate of molecular evolution in CCHFV. The rates of evolution, expressed as the number of substitutions per site per year and estimated across various studies, are shown. Each estimation includes the 95% highest posterior density interval (HPDI) or the HPDI is not available (*NA*).

Genomic Segment	Rate of Evolution (Substitutions/Site/Year)	95% HPDI	References
S	1.09 × 10^−4^	0.17 × 10^−4^–2.09 × 10^−4^	[22]
	0.34 × 10^−4^	0–1.22 × 10^−4^	[61]
	0.60 × 10^−4^	*NA*	[39]
	1.30 × 10^−4^	0.62 × 10^−4^–2.00 × 10^−4^	[59]
M	1.52 × 10^−4^	0.62 × 10^−4^–2.40 × 10^−4^	[22]
	1.22 × 10^−4^	0−1.97 × 10^−4^	[61]
	0.92 × 10^−4^	*NA*	[39]
	1.00 × 10^−4^	0.65 × 10^−4^–1.40 × 10^−4^	[59]
L	0.58 × 10^−4^	0.15 × 10^−4^–1.03 × 10^−4^	[22]
	1.01 × 10^−4^	0.01 × 10^−4^–1.54 × 10^−4^	[61]
	0.64 × 10^−4^	*NA*	[39]
	0.80 × 10^−4^	0.60 × 10^−4^–1.10 × 10^−4^	[59]

Several studies showed that the M segment evolves faster than the S and L segments, which is consistent with its higher genetic diversity [22,39,41,59,61], likely driven by selective pressures to enhance host interactions and evade immune responses, particularly through the induction of neutralizing antibodies [33,60,61]. Actually, high-frequency mutations were identified in the glycoproteins Gn and Gc, as well as in NP, potentially affecting host–virus interactions [11,28,31]. In contrast, the L segment evolves at a rate comparable to, or slightly higher than, that of the S segment, though this varies among strains [39,59,61,80]. Stronger purifying selection appears to act on these segments. For example, certain mutations were found to affect RdRP stability and activity (Table 4), and some may be selected against due to the importance of maintaining the activity of this protein, which is essential for viral replication [6]. Additionally, no significant activity-altering mutations were reported in the OTU protease domain [71]. We noted that further studies are necessary to assess potential evolutionary accelerations and decelerations that may occur in specific CCHFV genomic regions.

### 3.4. The Importance of Molecular Adaptation and Selection Along the CCHFV Genome

Molecular adaptation in CCHFV was broadly investigated using the traditional nonsynonymous/synonymous substitution rate ratio (*dN/dS*), which indicates genetic signatures of positive (diversifying), neutral, and negative (purifying) selection [89,90].

Interestingly, CCHFV exhibits different selective pressures in ticks compared to mammals. In a comparative study by Xia et al. (2016) [11], molecular adaptation of CCHFV was examined in ticks and mice (as mammalian representatives). The results showed that most functional domains evolve under purifying selection in mammals, where harmful mutations are generally eliminated. In contrast, CCHFV in ticks displayed a broader spectrum of selection pressures, with both purifying and diversifying selection detected across multiple genomic regions, suggesting a more dynamic evolutionary pattern in this host. Notably, the gene encoding the non-structural protein NSm exhibited strong signatures of positive selection in viruses collected from ticks, highlighting its potential in increasing amino acid diversity to promote virus–vector adaptation and transmission [11]. Selective pressures also varied among the three genomic segments but overall revealed a purifying selection pattern [91], consistent with the virus’s need to maintain essential protein activities despite its high mutation rate. Interestingly, variation in CCHFV selective pressure among tick species (specifically *Hyalomma* and *Rhipicephalus* genus) was reported, although both showed strong purifying selection and only small differences in the observed *dN/dS* ratios [92].

Selection in CCHFV protein-coding regions was also investigated at the local (codon) level. Most of the significantly selected sites were under purifying selection [92], once again indicating the importance of maintaining essential protein functions. However, some positively selected codon sites (PSSs) were identified. For example, three PSSs in the RdRp (positions 2011, 2438, and 2632) that were associated with potential effects in host adaptation and transmission were found to be established in specific viral lineages and geographic regions [91]. Similarly, five PSSs in the M segment (positions 13, 176, 368, 728, and 762) also showed geographically structured variation [91], as did a PSS in the L segment of CCHFV in *Rhipicephalus* ticks [92], supporting site-specific adaptations in particular viral lineages and host species.

In all, these findings indicate that, although the overall evolutionary pressure on the CCHFV genome is predominantly purifying, certain regions (such as RdRp and envelope glycoproteins) undergo local adaptations that can enhance transmission and immune evasion across different host species. This supports the hypothesis that, while positive selection is relatively rare, it can be important in fine-tuning viral protein functions. We believe that further studies are necessary to better understand the phenotypic consequences of the observed selection pressures and to predict potential molecular adaptations of CCHFV as it continues to expand into new hosts.

## 4. The Phylogeography of CCHFV and Its Rapid Geographic Expansion

The phylogeographic study of CCHFV is challenging not only due to limited genomic data but also because of the uneven geographic representation among available sequences. In particular, a higher number of sequences were obtained from Europe than from Asia and Africa [59]. Additionally, the phylogeographic distribution of CCHFV differs among its genomic segments as a result of segment reassortments [66,79,93], which were even detected in human patients [94]. Given that recombination and genomic reassortments can influence phylogenetic reconstructions [73,74], we recommend either a focus on non-reassortant CCHFV segments to reduce biases or the application of phylogenetic network approaches that account for such reticulate evolutionary events [95].

The S segment of CCHFV was classified into five to seven major lineages that include the genotypes Africa 1, Africa 2, Africa 3, Asia 1, Asia 2, Europe 1, and Europe 2, corresponding to strains from sub-Saharan Africa, Asia/the Middle East, and Europe (illustrated in Figure 3) [22,23,24,66]. Notably, the Europe 2 genotype (with strains from Greece) was later considered a distinct virus species (Aigai virus) due to its deep genetic divergence in the S segment and unique epidemiological features [96]. Regarding the M segment, phylogeographic analyses proposed up to seven lineages, which include a variety of genotypes, namely, Africa 1, Africa 2, Africa 3, Europe 1, Europe 2, Asia 1, and Asia 2, along with a distinct genotype found in Mauritania (illustrated in Figure 3) [22]. Indeed, studies based on the L segment also proposed several lineages, including African, Asian, and European genotypes (illustrated in Figure 3) [22]. This broad geographic distribution of certain viral clusters suggests long-distance migration of CCHFV variants.

In this regard, the rapid expansion of CCHFV lineages appears to be driven by a combination of environmental, ecological, and anthropogenic factors. Climate change is expanding the suitable habitats of *Hyalomma* tick vectors, allowing the virus to spread toward higher latitudes [97,98]. Specifically, warmer temperatures and altered rainfall patterns promote tick survival and establishment in areas that were previously unsuitable due to colder climates. In this regard, forecasting models also predict continued northward expansion of the virus [10,98]. At the same time, anthropogenic changes in land use (such as agricultural abandonment, habitat fragmentation, and an increase in wildlife host populations) can favor tick proliferation [29]. Migratory birds facilitate long-distance dispersal of viruses, promoting admixture between different viral populations and contributing to increased genetic diversity [99]. For instance, phylogeographic studies suggested that genotype 3 was likely introduced into Spain via migratory birds [14,100]. Similarly, livestock trade and animal movements also contribute to CCHFV expansion. For instance, the first reported CCHF cases in certain regions of Pakistan were associated with the arrival of sheep and goats, and other outbreaks were related to animal trade during Eid-ul-Adha [29,101]. In Europe, the arrival of genotype 5 into Spain in 2018 was hypothesized to be related to the importation of infected animals from Eastern Europe [100]. We believe that these multiple drivers of CCHFV geographic expansion, combined with frequent segment reassortments among viruses of different genotypes, are likely to further complicate the global phylogeographic structure of this virus, probably requiring the application of phylogenetic network reconstruction approaches to study it accurately. Indeed, the rapid evolutionary dynamics and intricate phylogeographic patterns of CCHFV, combined with the severity of its associated disease, underscore the importance of sustained genomic surveillance.

## 5. The Development of Effective Therapies Against CCHFV Is Influenced by the Virus’s Molecular Evolution

Based on experiences with other RNA viruses (e.g., influenza, hepatitis C, and SARS-CoV-2, among others), understanding viral evolutionary mechanisms can provide valuable insights for designing effective therapies [102,103]. For instance, identifying molecular targets that are relatively conserved and essential for viral replication is fundamental for the development of drugs and vaccines [104]. In the case of CCHFV, currently, there are no specific antiviral drugs or vaccines approved for its treatment. Ribavirin, a broad-spectrum antiviral drug that interferes with RNA synthesis, is commonly used, although its efficacy was not demonstrated [105]. Diverse antiviral compounds are under development, but they face the challenge of targeting continuously evolving viral strains derived from the rapid genetic diversification of CCHFV [106]. Intense efforts are focused on vaccine design. A variety of vaccine strategies are currently under investigation, including plant-expressed vaccines, inactivated vaccines, viral vector vaccines, subunit vaccines, virus-like replicon particle vaccines, DNA vaccines, and mRNA vaccines; also, genetic adjuvants are being explored to enhance immune responses, among other approaches [107]. However, despite the numerous vaccine candidates, the development of an efficient vaccine remains challenging. A described technical limitation is the lack of suitable animal models that accurately replicate the pathology of the virus in human infections [107,108]. In addition, beyond multiple technical factors (i.e., accounting for vaccine antigenicity, allergenicity, toxicity, water solubility, and stability), an important obstacle is again the virus’s rapid and multifactorial evolution (driven by frequent mutations, recombination events, and genomic reassortments). These processes can lead to the emergence of novel strains that may reduce vaccine coverage across viral populations or even completely evade immune recognition. Actually, in some preclinical evaluations of potential vaccines, the protective efficacy of vaccine candidates against genetically distant viral variants was not adequately assessed. In this context, antigen selection and epitope prediction, which are standard steps in vaccine development, require a comprehensive understanding of the virus’s molecular evolution in order to preserve vaccine efficacy and immune recognition over time.

The glycoproteins Gn and Gc, along with the NP, have been identified as key antigenic targets in nearly all CCHFV vaccine studies. Regarding Gn and Gc, they are generally considered preferred antigens, as they are located in the viral envelope and can elicit neutralizing antibodies [109,110,111,112]. However, identifying specific protective epitopes remains challenging due to the rapid evolution of these proteins (as previously noted, the M segment encoding them is highly variable among strains) and the still limited evaluation of their immune properties [65,107]. Nevertheless, various vaccine platforms have employed GPC-derived constructs, demonstrating that these proteins can induce certain neutralizing responses in mice [113,114]. In contrast, the NP evolves at a relatively slower rate than the coding regions of the M and L segments, making it a promising target for broader population coverage, which has led to its widespread exploration in vaccine development [108,111,115]. Actually, it is considered a dominant immunogen in CCHFV and other related bunyaviruses, capable of eliciting both humoral and cellular responses [116,117]. Despite its relatively high conservation, NP-based vaccines exhibited varying protective efficacy when used alone [108,116], which depended on the platforms used, and showed high protection in some promising vaccines [118,119]. Combining NP with structural proteins such as Gn and Gc showed improved protection, especially when delivered through viral vectors or virus-like particles [108,120,121], although the exact contribution of each antigen to protective immunity remains to be elucidated. In general, the lack of clarity regarding the roles of individual viral components and host immune factors continues to hinder rational vaccine design. Another potential target is the region of the M segment encoding non-structural proteins, which, due to its variability, could potentially help address the genetic heterogeneity of circulating viral strains [122,123]. The idea is that incorporating a comprehensive understanding of CCHFV molecular evolution and host adaptation could inform the rational design of vaccines with broad-spectrum efficacy against genetically diverse viral strains.

Current platforms, such as viral vector vaccines [110,124], DNA vaccines [125], mRNA vaccines [126], or virus-like particle vaccines [127], have shown varying levels of success but often ignore the extensive genetic diversity of CCHFV circulating strains [108,110]. The over-reliance on homologous testing represents a significant limitation, as it may lead to an overestimation of vaccine efficacy. Therefore, evaluations of heterologous protection are needed. In fact, vaccine candidates expressing the same GPC showed protection in some conditions [120,128] while failing in others, depending on the platform and strains used [107,108]. These inconsistencies should be addressed in order to advance the development of effective CCHFV vaccines. We believe that the development of efficient CCHFV vaccines requires interdisciplinary collaboration, combining virology, immunology, and evolutionary biology, supported by appropriate animal models and guided and tested by accounting for the virus’s evolutionary dynamics and derived genetic diversity.

## 6. Conclusions and Future Prospects

CCHFV represents a significant global health threat due to its high fatality rate and extensive geographic distribution, which continues to expand. Indeed, this virus exhibits a strong capacity for rapid evolution through mutations, recombination events, and genomic reassortments, leading to high genetic diversity that contributes to complicating the development of broadly effective treatments and vaccines. CCHFV will probably continue to spread geographically and increase in genetic diversity. Whether emerging variants will affect its transmissibility, the virulence of the associate disease, or the development and effectiveness of therapies is difficult to predict, given the complex evolutionary dynamics of CCHFV. However, monitoring and understanding its molecular evolution can provide valuable insights to anticipate and respond in unfavorable scenarios. Ideally, if the virus’s evolutionary patterns can be understood and modeled, approaches for forecasting viral evolution could be developed to anticipate the emergence of new strains [129,130,131,132], which could facilitate the development of effective and durable antiviral treatments and vaccines as well as help anticipate future needs. Therefore, we believe that continuous surveillance of CCHFV, with close attention to its evolutionary processes and emerging variants, is crucial for anticipating and mitigating potential outbreaks and improving public health responses.

## Figures and Tables

**Figure 1 viruses-17-01054-f001:**
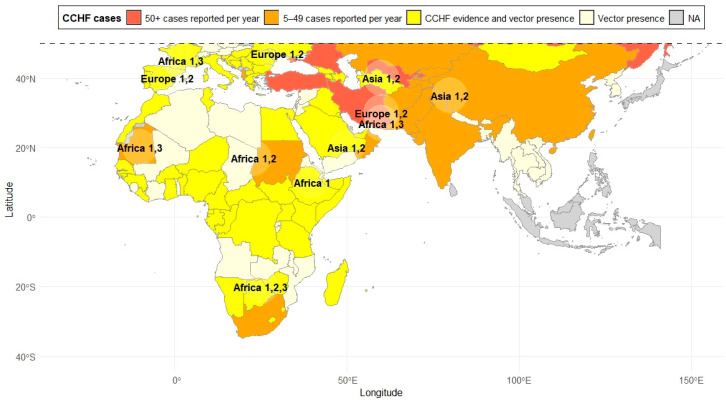
Geographic distribution of CCHF cases and CCHFV genotypes. Cases are shown in color, while the genotypes are indicated on the map. Data obtained from various previous studies [5,6,12,13,15,16].

**Figure 2 viruses-17-01054-f002:**
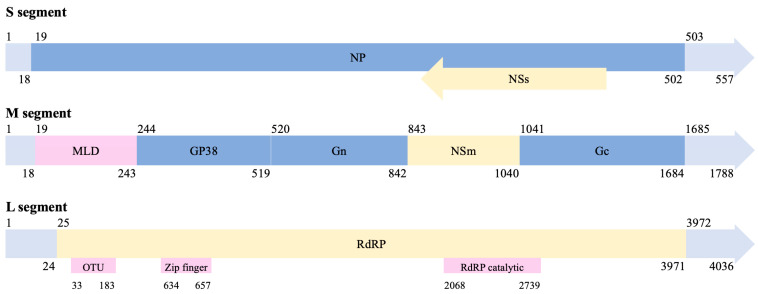
The CCHFV genome structure. Illustrative representation of the CCHFV genomic regions according to the GenBank references NC_005302 [30] (S segment), NC_005300 [31] (M segment), and NC_005301 [32] (L segment). Non-coding regions are shown in clear blue, local domains are shown in purple, regions encoding non-structural proteins are shown in yellow, and regions encoding structural proteins are shown in dark blue. The S segment includes the nucleoprotein (NP) and the non-structural protein (NSs). The M segment includes the mucin-like domain (MLD), glycoprotein 38 (GP38), glycoproteins N and C (Gn and Gc, respectively), and the non-structural protein (NSm). The L segment contains the RNA-dependent RNA polymerase (RdRP), which includes the OTU protease domain (OTU), the Zip finger domain (Zip finger), and the RdRP catalytic domain (RdRP catalytic). The numbers indicated in the figure refer to region lengths based on nucleotide triplets, and should be considered as approximations since possible insertion and deletion events, among others processes, can affect region lengths. Indeed, the position of each amino acid mutation cited along the manuscript is based on the associated reference, and may differ among articles and ranges specified in this illustrative representation.

**Figure 3 viruses-17-01054-f003:**
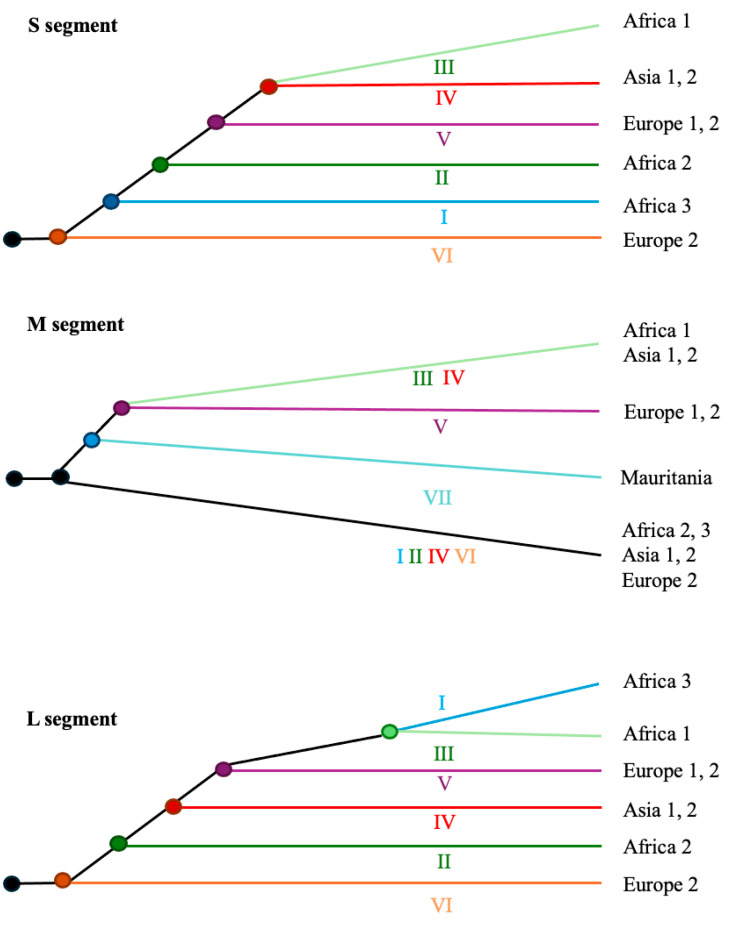
General evolutionary history of the main CCHFV lineages and genotypes for each genomic segment. Illustrative representation of the phylogenetic relationships among the main CCHFV lineages and genotypes for each genomic segment [6,22]. Viral lineages are labeled with Roman numerals (and can include one or more genotypes associated with geographic regions) as follows: I (Africa 3: South Africa and West Africa), II (Africa 2: Democratic Republic of the Congo), III (Africa 1: West Africa), IV (Asia 1 and 2: Asia and the Middle East), V (Europe 1 and 2: Europe and Turkey), VI (Europe 2: Greece), and VII (Mauritania).

## Data Availability

Not applicable.

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
