# Peer review of "Molecular Evolution and Phylogeography of the Crimean–Congo Hemorrhagic Fever Virus"

_viruses, 2025, doi:10.3390/v17081054_

Round 1
Reviewer 1 Report
Comments and Suggestions for Authors
The review paper was interesting and a good summary of the current state of CCHFV lineages, potentially important mutations, and important considerations for developing universal CCHFV vaccines. I have not seen a good summary of current research describing how various mutations in CCHFV proteins may affect viral function and it was exciting to see the authors putting this data together in a concise and easy to follow table. This was a hole in the current CCHFV literature and an important contribute to allowing researchers to begin identifying patterns of mutations that could greatly impact therapeutic development or understanding of CCHFV pathogenesis.
Major edit suggestions:
Table 2 – mutation K90A “reduces functionality” is too vague; mutations I228M and K251E can report as “potentially increases virulence in mice” or something similar, S301G can report as “may affect rigidity of protein” or something similar
Table 3 and Table 4 – as with suggestions for Table 2, the authors should fill in the potential impacts that these mutations may have according to the papers they are reported in to make the table more impactful and allow readers to identify any patterns in potential functions of mutations, this is in reference to all mutations with “-“ as the influence on protein function.
Minor edit suggestions:
Figure 1 – switch the order of the key colors so that the red “50+” cases is all the way to the left, followed by the orange then yellow and so on
Line 65 – references a hantavirus paper which mentions CCHFV and references several papers about CCHFV recombination, the authors should find and reference the original source material and not the hantavirus paper
Line 73-77 – is there a reference for the subsequent studies?
Figure 2 – is there any meaning to the different coloring on the genome?
Lines 73-77/Figure 3 – the text says 7 S segment lineages, 7 M segment, and 6 L segment but in the figure, there are 6 for S and 4 for M? perhaps I am understanding the figure incorrectly, authors could clarify in the legend
Lines 118-119 – formatting fix “Additionally, the…Nucleoprotein” is split into two paragraphs
Line 129-131 – add reference for MxA information
Line 157 – reference for GP38 contributing to virion assembly is incorrect (ref 44)
Line 172 – type “OUT” should be “OTU”
Line 186-188 – may include the citation for Hawman et al. Immunocompetent mouse model (ref. 125), this mouse-adapted CCHFV includes a mutation in the NP which may support the claim that mutations in NP affect RNA binding, immune evasion (this mutation may be what enabled CCHFV to become mouse-adapted)
Section lines 191-206, can you describe the specifics of the variability in the M segment? e.g. is this variability in the Gn/Gc or across the non-structural proteins as this may impact what we se in terms of CCHFV infectivity variability. The MLD is also known to be highly variable, and it is worth noting if this is what accounts for the high variability or not.
Line 208 – as above may reference Hawman et al. Immunocompetent mouse model (ref. 125) as this supports the claim that mutations in key proteins affect virus-host interactions, specifically infectivity here
Line 253 – should the references be at the end of the sentence not in the middle?
Line 276 – ref. 76 is an influenza reference?
Line 355-356 – should “focusing” be “focus” and “applying” to “apply”
Line 379 – remove “to” from the sentence
Line 380-381 – that ref. 92 is not directly talking about CCHFV so perhaps make this sentence more general referencing “viruses” not “the virus”
Line 401-402 – says “currently” twice, remove one of them for better sentence flow
Line 428 – extra space before “however?”
Line 438-439 – describes inconsistent efficacy when NP is used alone however this depends on the platform, some NP-only vaccines are highly protective (see publications https://doi.org/10.1016/j.ebiom.2025.105698, https://doi.org/10.1016/j.ebiom.2022.104188) this may be worth noting to give a better overview of the current state of the cchfv vaccine field. Also relevant as these were tested against a heterologous challenge strain
Line 467 – “complicating” not “complicate?”
Unclear if the references are in the correct order? The last references of the paper appear to be ref 120-123 but the reference list goes up to 127?
Author Response
Reviewer #1
The review paper was interesting and a good summary of the current state of CCHFV lineages, potentially important mutations, and important considerations for developing universal CCHFV vaccines. I have not seen a good summary of current research describing how various mutations in CCHFV proteins may affect viral function and it was exciting to see the authors putting this data together in a concise and easy to follow table. This was a hole in the current CCHFV literature and an important contribute to allowing researchers to begin identifying patterns of mutations that could greatly impact therapeutic development or understanding of CCHFV pathogenesis.
AU: We thank the reviewer for the positive comments on our work. In this revised version, we have addressed the concerns raised below. We have now provided more information about the functional role of some mutations when available (note that the specific effect of certain mutations is not clear yet in the CCHFV literature). Indeed, we have incorporated all the other suggestions.
Major edit suggestions:
Table 2 – mutation K90A “reduces functionality” is too vague; mutations I228M and K251E can report as “potentially increases virulence in mice” or something similar, S301G can report as “may affect rigidity of protein” or something similar
Table 3 and Table 4 – as with suggestions for Table 2, the authors should fill in the potential impacts that these mutations may have according to the papers they are reported in to make the table more impactful and allow readers to identify any patterns in potential functions of mutations, this is in reference to all mutations with “-“ as the influence on protein function.
AU: We have now further revised the literature for each mutation indicated in tables 2-4 and updated these tables.
Minor edit suggestions:
Figure 1 – switch the order of the key colors so that the red “50+” cases is all the way to the left, followed by the orange then yellow and so on
AU: We agree, fixed.
Line 65 – references a hantavirus paper which mentions CCHFV and references several papers about CCHFV recombination, the authors should find and reference the original source material and not the hantavirus paper
AU: Thanks for indicating this error. Fixed.
Line 73-77 – is there a reference for the subsequent studies?
AU: Added.
Figure 2 – is there any meaning to the different coloring on the genome?
AU: We have now revised the colors of the coding regions and included in the figure caption their meaning. In particular, non-coding regions are shown in clear blue, local domains are now shown in purple, regions encoding non-structural proteins are now shown in yellow, and regions encoding structural proteins are now shown in dark blue.
Lines 73-77/Figure 3 – the text says 7 S segment lineages, 7 M segment, and 6 L segment but in the figure, there are 6 for S and 4 for M? perhaps I am understanding the figure incorrectly, authors could clarify in the legend
AU: We thank for this comment because it helped us to identify a confusion in terminology in those lines of the manuscript. The S segment has 6 lineages (shown in Figure 3 using Roman numerals) and 7 genotypes. The M segment has 7 lineages and 8 genotypes and, the L segment included 6 lineages and 7 genotypes. Note that terminal branches in the provided phylogenies can include one or more than one lineage (Roman numerals) and genotype (associated to geographic regions). We have now clarified this distinction in both the main text and the caption of Figure 3.
Lines 118-119 – formatting fix “Additionally, the…Nucleoprotein” is split into two paragraphs
AU: Fixed.
Line 129-131 – add reference for MxA information
AU: Added.
Line 157 – reference for GP38 contributing to virion assembly is incorrect (ref 44)
AU: We agree, changed.
Line 172 – type “OUT” should be “OTU”
AU: Fixed.
Line 186-188 – may include the citation for Hawman et al. Immunocompetent mouse model (ref. 125), this mouse-adapted CCHFV includes a mutation in the NP which may support the claim that mutations in NP affect RNA binding, immune evasion (this mutation may be what enabled CCHFV to become mouse-adapted)
AU: Thanks, added.
Section lines 191-206, can you describe the specifics of the variability in the M segment? e.g. is this variability in the Gn/Gc or across the non-structural proteins as this may impact what we se in terms of CCHFV infectivity variability. The MLD is also known to be highly variable, and it is worth noting if this is what accounts for the high variability or not.
AU: This is an interesting aspect. Indeed, MLD typically shows the highest sequence variability among CCHFV proteins. The envelope glycoproteins Gn and Gc also display high diversity, generally lower than that of the MLD, but higher than that observed for the NS protein. We have now included this information in the manuscript, including references that support it.
Line 208 – as above may reference Hawman et al. Immunocompetent mouse model (ref. 125) as this supports the claim that mutations in key proteins affect virus-host interactions, specifically infectivity here
AU: Thanks, added.
Line 253 – should the references be at the end of the sentence not in the middle?
AU: Fixed.
Line 276 – ref. 76 is an influenza reference?
AU: Thanks for indicating us this error. Changed.
Line 355-356 – should “focusing” be “focus” and “applying” to “apply”
AU: Yes, fixed.
Line 379 – remove “to” from the sentence
AU: Fixed.
Line 380-381 – that ref. 92 is not directly talking about CCHFV so perhaps make this sentence more general referencing “viruses” not “the virus”
AU: We agree, changed.
Line 401-402 – says “currently” twice, remove one of them for better sentence flow
AU: Fixed.
Line 428 – extra space before “however?”
AU: Fixed.
Line 438-439 – describes inconsistent efficacy when NP is used alone however this depends on the platform, some NP-only vaccines are highly protective (see publications https://doi.org/10.1016/j.ebiom.2025.105698, https://doi.org/10.1016/j.ebiom.2022.104188) this may be worth noting to give a better overview of the current state of the cchfv vaccine field. Also relevant as these were tested against a heterologous challenge strain
AU: Thanks for indicating this. We have now revised this sentence to clarify that the generated protection was high in some platforms, and it was supported with references.
Line 467 – “complicating” not “complicate?”
AU: Fixed.
Unclear if the references are in the correct order? The last references of the paper appear to be ref 120-123 but the reference list goes up to 127?
AU: The journal automatically reformatted our originally submitted manuscript by moving the tables and figures (and its captions) into the corresponding sections of the text. This change affected the order of the references in the version that was sent to the reviewers. For example, reference 127 (the last one) is cited in the caption of Figure 1 (figure captions were placed at the end of the manuscript in our original version of the manuscript). This formatting change also caused some of the points raised by the reviewer. We have now submitted the manuscript in the journal format, with the references properly ordered and other formatting issues resolved.

Reviewer 2 Report
Comments and Suggestions for Authors
This review article entitled ‘’Molecular evolution and phylogeography of the Crimean-Congo Hemorrhagic Fever Virus’’ presents a comprehensive review regarding the status of CCHF virüs. It provides a brief overview of comprehensive compilation of information on the molecular and phylogeographic evaluation of the virus, addressing the geographic distribution of CCHF cases and CCHFV genotypes occure in the World. The main objectives are to provide a brief molecular evolution, genetic characterization and phylogeography of CCHFV, and discuss their potential implications for therapeutic design. Specifically, it focus on the description of the virus capacity to increase its genetic diversity through numerous mutations, recombination events, and genomic reassortments, which affect fundamental viral functions such as RNA binding. The rewiev also provides important informations regarding the host-virus interactions, viral entry, and polymerase activity. As a result, it is a comprehensive, well-organized, and well-written review article. Appropriate referencing and a thorough review of the current status basic virological information of CCHF virüs.
Comments:
I have a minor just only seggestion that:
It is well konow that there a no vaccine against the dieseae, but preclinical vaccine studies for various vaccine platforms have shown promising results. However, the genetic variability of CCHFV makes it challenging to develop a vaccine that can provide broad protection against all strains of the virus. This comprehensive review could have included a section on vaccination?
Author Response
Reviewer #2
This review article entitled ‘’Molecular evolution and phylogeography of the Crimean-Congo Hemorrhagic Fever Virus’’ presents a comprehensive review regarding the status of CCHF virüs. It provides a brief overview of comprehensive compilation of information on the molecular and phylogeographic evaluation of the virus, addressing the geographic distribution of CCHF cases and CCHFV genotypes occure in the World. The main objectives are to provide a brief molecular evolution, genetic characterization and phylogeography of CCHFV, and discuss their potential implications for therapeutic design. Specifically, it focus on the description of the virus capacity to increase its genetic diversity through numerous mutations, recombination events, and genomic reassortments, which affect fundamental viral functions such as RNA binding. The rewiev also provides important informations regarding the host-virus interactions, viral entry, and polymerase activity. As a result, it is a comprehensive, well-organized, and well-written review article. Appropriate referencing and a thorough review of the current status basic virological information of CCHF virus.
AU: We thank the reviewer for the positive comments on our work.
Comments:
I have a minor just only seggestion that:
It is well konow that there a no vaccine against the dieseae, but preclinical vaccine studies for various vaccine platforms have shown promising results. However, the genetic variability of CCHFV makes it challenging to develop a vaccine that can provide broad protection against all strains of the virus. This comprehensive review could have included a section on vaccination?
AU: The reviewer is right in pointing out that diverse preclinical vaccines are currently underway and that the genetic variability of CCHFV poses a major challenge for vaccine development. We have included these aspects in the section “The development of effective therapies against CCHFV is influenced by the virus molecular evolution”, which includes the design and development of therapies against CCHFV from the perspective of accounting for the virus molecular evolution, highlighting the importance of understanding CCHFV molecular evolution in the context of therapeutic development. Regarding the suggestion of adding a more detailed and technical section on CCHFV vaccines, we believe that this would fall outside the scope of the present review. As indicated in the title and abstract, this review focuses on the molecular evolution and genetic variability of CCHFV, which are fundamental for this virus. Adding additional areas of research of CCHFV such as detailed state of vaccines development, viral replication mechanisms or immunology, would significantly broaden the scope and could lead to a journal issue or a book. Nevertheless, we have cited some relevant reviews that cover those complementary topics. In any case, we appreciate the suggestion of the reviewer.
